# A Probabilistic Bag-to-Class Approach to Multiple-Instance Learning

**Kajsa Møllersen** [1,*] **, Jon Yngve Hardeberg** [2] **and Fred Godtliebsen** [3]

[1] Department of Community Medicine, Faculty of Health Science, UiT The Arctic University of Norway, N-9037 Tromsø, Norway

[2] Department of Computer Science, Faculty of Information Technology and Electrical Engineering, NTNU—Norwegian University of Science and Technology, N-2815 Gjøvik, Norway; jon.hardeberg@ntnu.no

[3] Department of Mathematics and Statistics, Faculty of Science and Technology, UiT The Arctic University of Norway, N-9037 Tromsø, Norway; fred.godtliebsen@uit.no

* Correspondence: kajsa.mollersen@uit.no; Tel.: +47-9778-3940

**Abstract:** Multi-instance (MI) learning is a branch of machine learning, where each object (bag) consists of multiple feature vectors (instances)—for example, an image consisting of multiple patches and their corresponding feature vectors. In MI classification, each bag in the training set has a class label, but the instances are unlabeled. The instances are most commonly regarded as a set of points in a multi-dimensional space. Alternatively, instances are viewed as realizations of random vectors with corresponding probability distribution, where the bag is the distribution, not the realizations. By introducing the probability distribution space to bag-level classification problems, dissimilarities between probility distributions (divergences) can be applied. The bag-to-bag Kullback–Leibler information is asymptotically the best classifier, but the typical sparseness of MI training sets is an obstacle. We introduce bag-to-class divergence to MI learning, emphasizing the hierarchical nature of the random vectors that makes bags from the same class different. We propose two properties for bag-to-class divergences, and an additional property for sparse training sets, and propose a dissimilarity measure that fulfils them. Its performance is demonstrated on synthetic and real data. The probability distribution space is valid for MI learning, both for the theoretical analysis and applications.

**Dataset:** Breast tissue images available at https://bioimage.ucsb.edu/research/bio-segmentation, extracted feature vectors available at https://figshare.com/articles/MIProblems_A_repository_of_multiple_instance_learning_datasets/6633983. *BreakHis* data available at https://web.inf.ufpr.br/vri/databases/breast-cancer-histopathological-database-breakhis/. Code available at https://github.com/kajsam/ProbabilisticBag2Class.

**Keywords:** image classification; multi-instance learning; divergence; dissimilarity; bag-to-class; Kullback–Leibler

## 1. Introduction

### 1.1. Classification of Weakly Supervised Data

Machine-learning applications include a wide variety of data types, images being one of the most successful areas. It has had an enormous impact on image analysis, especially in replacing small sets of hand-crafted features with large sets of computer readable features, which often lack apparent

or intuitive meaning. The task and problems to which machine learning is applied can be divided broadly into unsupervised and supervised learning. In supervised learning, the training data consists of $K$ objects, $\mathbf{x}$, with corresponding class labels, $y$; $\{(\mathbf{x}_1, y_1), \ldots, (\mathbf{x}_k, y_k), \ldots, (\mathbf{x}_K, y_K)\}$. An object is typically a vector of $d$ feature values, $\mathbf{x}_k = (x_{k1}, \ldots, x_{kd})$, observed directly or extracted from e.g., an image. In classification, the task is to build a classifier that correctly labels a new object. The training data is used to adjust the model according to the desired outcome, often maximizing the accuracy of the classifier.

For many types of images, only a small part of the image defines the class, but the label is available only at image level. This is common in medical images, such as histology slides, where the tumor cells typically make up a small proportion of the image. However, the location of those cells is not available for training. Multi-instance (MI) learning is a branch of machine learning that specifically targets problems where labels are available only at a superior level, and relates to other weakly supervised data problems, such as semi-supervised learning and transfer learning through label scarcity [1].

## 1.2. Multi-Instance Learning

In MI learning, each object is a set of feature vectors referred to as instances. The set $\mathbb{X}_k = \{\mathbf{x}_{k1}, \ldots, \mathbf{x}_{kn_k}\}$, where the $n_k$ elements are vectors of length $d$, is referred to as bag. The number of instances, $n_k$, varies from bag to bag, whereas the vector length is constant. In supervised MI learning, the training data consists of $K$ sets and their corresponding class labels, $\{(\mathbb{X}_1, y_1), \ldots, (\mathbb{X}_k, y_k), \ldots, (\mathbb{X}_K, y_K)\}$.

Figure 1a shows an image (bag), $k$, of benign breast tissue [2], divided into $n_k$ segments with corresponding feature vectors (instances) $\mathbf{x}_{k1}, \ldots, \mathbf{x}_{kn_k}$ [3]. Correspondingly, Figure 1b shows malignant breast tissue.

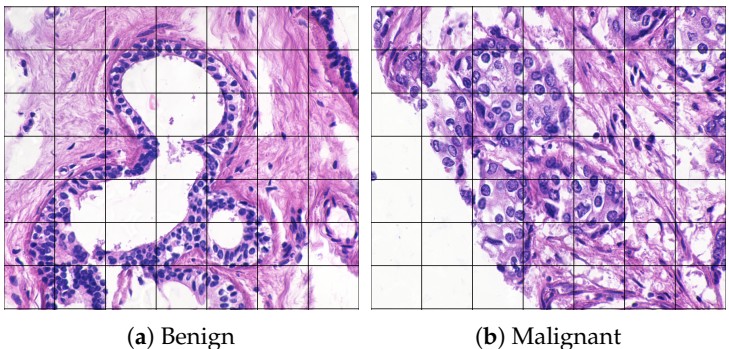

(**a**) Benign        (**b**) Malignant

**Figure 1.** Breast tissue images [2]. The image segments are not labeled.

The images in the data set have class labels; however, the individual segments do not. This is a key characteristic of MI learning—the instances are not labeled. MI learning includes instance classification [4], clustering [5], regression [5], and multi-label learning [6,7], but this article will focus on bag classification. MI learning can also be found as integrated parts of end-to-end methods for image analysis that generate patches, extract features and do feature selection [7]. See also [8] for an overview and discussion on end-to-end neural network MI learning methods.

The term "MI learning" was introduced in an application of molecules (bags) with different shapes (instances), and their ability to bind to other molecules [9]. A molecule binds if at least one of its shapes can bind. In MI terminology, the classes in binary classification are referred to as positive, *pos*, and negative, *neg*. The assumption that a positive bag contains at least one positive instance, and a negative bag contains only negative instances is referred to as the standard MI assumption.

Many new applications violate the standard MI assumption, such as image classification [10] and text categorization [11]. Consequently, successful algorithms meet more general assumptions, see e.g., the hierarchy of Weidmann et al. [12] or Foulds and Frank's taxonomy [13]. For a more recent



review of MI classification algorithms, see e.g., [14]. Amores [15] presented the three paradigms of instance space (IS), embedded space (ES), and bag space (BS). IS methods aggregate the outcome of single-instance classifiers applied to the instances of a bag, whereas ES methods map the instances to a vector, followed by use of a single-instance classifier. In the BS paradigm, the instances are transformed to a non-vectorial space where the classification is performed, avoiding the detour via single-instance classifiers. The non-vectorial space of probability functions has not yet been introduced to the BS paradigm, despite its analytical benefits, see Sections 3.2 and 3.3.

Although both Carbonneau et al. [16] and Amores [15] defined a bag as a set of feature vectors, Foulds and Frank [13] stated that a bag can also be modelled as a probability distribution. The distinction is necessary in analysis of classification approaches, and both viewpoints offer benefits, see Section 6.1 for a discussion.

### 1.3. Bag Density and Class Sparsity

Optimal classification in MI learning depends on the number of instances per bag (bag density) and the number of bags per class in the training set (class density). Sample sparsity is a common obstacle in MI learning [16], which we address in Section 3.5. High bag density ensures a precise description of each bag, whereas high class density ensures precise modelling of each class when training the classifier. In image analysis, the number of patches corresponds to the number of instances, and is commonly a user input parameter. The number of images corresponds to the number of bags, and is limited by the training set itself.

High resolution of today's images and the increasingly common practice of sharing the images themselves instead of extracted features ensure high bag density. The number of bags available for training is still limited, and will continue to be so in the foreseeable future, especially for medical images where data collection is restricted by laws and regulations. This motivates an approach to MI learning that can exploit the increasing bag density and overcome the class sparsity.

### 1.4. A Probabilistic Bag-to-Class Approach to Multi-Instance Learning

We propose to model the bags as probability distributions and the instances as random samples. The bags are assumed to be random samples from their respective classes and the instance-bag sampling form a hierarchical distribution. Hierarchical distribution is novel for bag classification and novel outside the strict standard MI assumption. Unbiased estimators for the bag probability distributions ensure that as the number of instances increases ($n_k \to \infty$), the discrepancy between the estimate and the underlying truth diminishes, taking advantage of increasing bag density. To overcome the problem of class sparsity, the instances are aggregated at class level.

We further propose to use a bag-to-class dissimilarity measure for classification. This is novel in the MI context, where dissimilarity measures have been either instance-to-instance or bag-to-bag. With the analytical framework of probability distributions and their dissimilarity measures, we present the optimal classifier for dense class sampling as a theoretical background and identify data-independent properties for bag classification under class sparsity.

The main contribution of this article is a bag-to-class dissimilarity measure for sparse training data. It builds on:

- presenting the hierarchical model for general, non-standard MI assumptions (Section 3.3),
- introduction of bag-to-class dissimilarity measures (Section 3.5), and
- identification of two properties for bag-to-class divergence (Section 4.1).

The novelty is that it takes into account the class sparsity by comparing a bag to one class while conditioning on the other class.

In Section 5, the Kullback–Leibler (KL) information and the new dissimilarity measure is applied to data sets and the results are reported. Bags defined in the probability distribution space, in combination

with bag-to-class divergence, constitutes a new framework for MI learning, which is compared to other frameworks in Section 6.

## 2. Related Work and State-of-the-Art

The feature vector set viewpoint seems to be the most common, but the probabilistic viewpoint was introduced already in 1998, under the assumption that instances of the same class are independent and identically distributed (i.i.d.) [17]. This assumption has been used in approaches such as estimating the expectation by the mean [18], or estimation of class distribution parameters [19], but has also been criticized [20]. The hierarchical distribution was introduced for learnability theory under the standard MI assumption for instance classification in 2016 [4]. We expand the use for more general assumptions in Section 3.3.

Dissimilarities in MI learning have been categorized as instance-to-instance or bag-to-bag [15,21]. The bag-to-prototype approach in [21] offers an in-between category, but the theoretical framework is missing. Bag-to-class dissimilarity has not been studied within the MI framework, but has been used under the i.i.d. given class assumption for image classification [22]. The sparseness of training sets was also addressed: if the instances are aggregated on class level, a denser representation is achieved. Many MI algorithms use dissimilarities, e.g., graph distances [23], Hausdorff metrics [24], functions of the Euclidean distance [14,25], and distribution parameter-based distances [14]. The performances of dissimilarities on specific data sets have been investigated [14,19,21,25,26], but more analytical comparisons are missing. A large class of commonly used kernels are also distances [27], and hence, many kernel-based approaches in MI learning can be viewed as dissimilarity-based approaches. In [28], the Fisher kernel is used as input to a support vector machine (SVM), whereas in [11,20] the kernels are an integrated part of the methods.

The non-vectorial graph space was used in [20,23]. We introduce the non-vectorial space of probability functions as an extension within the BS paradigm for bag classification through dissimilarity measures between distributions in Section 3.2.

The KL information was applied in [22], and is a much-used divergence function. It is closely connected to the Fisher information [29] used in [28] and to the cross entropy used as loss function in [8]. We propose a conditional KL information in Section 4.2, which differs from the earlier proposed weighted KL information [30] whose weight is a constant function of $X$.

There is a wide variety in MI learning, both in methods and data sets, and it should be clear that state-of-the-art will depend on the type of data. Sudharshan et al. [31] gave a comparison of 12 MI classification methods and five state-of-the-art general classification methods on a well-described, publicly available histology image data set. All methods included have shown best performance on other data sets. The five methods that showed best performance for at least one of the data subsets serve as state-of-the-art baseline for evaluation in Section 5.3.

Cheplygina et al. [1] gave an overview of MI learning applications in different categories, but no comparison was made. The work of Sudharshan et al. falls into the "Histology/Microscopy" category, and the overview offers a potential expansion of histology state-of-the-art. Among the 12 listed articles, Zhang et al. [32] concluded that GPMIL outperforms Citation-kNN, which is one of the 12 methods in [31], but not one of the 5 best-performing. Kandemir et al. [3], Li et al. [33] and Tomczak et al. [34] presented methods that outperform GPMIL on a publicly available data set. We include these as comparison.

Of the remaining articles, none of them present an extensive comparison to other methods, their data sets are either non-public [35–38], no longer available [39], or the reference is not complete [40,41], which make them unsuitable for comparison. Jia et al. [42] presented a segmentation method, and is therefore not comparable.

## 3. Theoretical Background and Intuitions

### 3.1. Notation

Subscript and superscript *pos* and *neg* refer to the class label of the bag, subscript and superscript $+$ and $-$ refer to the unknown instance label.

$X$ : instance random vector
$C$ : class, either *pos* or *neg*
$B$ : bag
$P(\cdot)$ : probability distribution
$\mathbf{x}_{ki}$ : feature vector (instance) in set $k$, $i = 1, \ldots, n_k$
$\mathbb{X}_k$: set of feature vectors $k$ of size $n_k$
$y_k$: bag label
$\mathcal{X}$ : sample space for instances
$\mathcal{X}^+$ : sample space for positive instances
$\mathcal{X}^-$ : sample space for negative instances
$\mathcal{B}_{pos}$ : sample space of positive bags
$\mathcal{B}_{neg}$ : sample space of negative bags
$P(C|\mathbb{X}_k)$ : posterior class probability, given instance sample
$\Theta$ : parameter random vector
$\theta_k$ : parameter vector
$P_{bag}(X) = P(X|B)$ : probability distribution for instances in bag $B$
$P(X|\theta_k)$ : parameterized probability distribution of bag $k$
$P_{pos}(X) = P(X|pos)$ : probability distribution for instances from the positive class
$P_{neg}(X) = P(X|neg)$ : probability distribution for instances from the negative class
$\tau_i$ : instance label
$\pi_k$ : probability of positive instances
$D(P_k, P_\ell) = D(P_k(X), P_\ell(X))$: divergence from $P_k(X)$ to $P_\ell(X)$
$f_k(\mathbf{x}) = f(\mathbf{x}|\theta_k)$ : probability density function (PDF) for bag $k$
$D(f_k, f_\ell) = D(f_k(\mathbf{x}), f_\ell(\mathbf{x}))$: divergence from $f_k(\mathbf{x})$ to $f_\ell(\mathbf{x})$

We assume $P(X) < \infty$, and equivalently $f(\mathbf{x}) < \infty$, for all distributions.

### 3.2. The Non-Vectorial Space of Probability Functions

The intuition behind the probabilistic approach in MI learning can be understood through image analysis and tumor classification. Figure 1a represents parts of a tumor, chosen carefully for diagnostic purposes. The process from biological material to image contains steps whose outcome is influenced by subjective choices and randomness: The precise day the patient is admitted influences the state of the tumor; the specific parts of the tumor that are extracted for staining; the actual stain varies from batch to batch, and the imaging equipment has multiple parameter settings. All this means that the same tumor would have produced a different image under different circumstances. The process from image to feature vector set also contains several steps: Patch size, grid or random patches, color conversion, etc. In summary, the observed feature vectors are a representation of an underlying object, and that representation may vary, even if the object remains fixed.

From the probabilistic viewpoint, an instance, $\mathbf{x}$, is a realization of a random vector, $X$, with probability distribution $P(X)$ and sample space $\mathcal{X}$. The bag is the probability distribution $P(X)$, and the set of instances, $\mathbb{X}$, is multiple realizations of $X$. The task of an MI classifier is to classify the bag given the observations, $\mathbb{X}$.

The posterior class probability, $P(C|\mathbb{X}_k)$, is an effective classifier if the standard MI assumption holds, since it is defined as:

$$P(pos|\mathbb{X}_k) = \begin{cases} 1 \text{ if any } \mathbf{x}_{ki} \in \mathcal{X}^+, i = 1, \ldots, n_k \\ 0 \text{ otherwise,} \end{cases}$$

where $\mathcal{X}^+$ is the positive instance space, and the positive and negative instance spaces are disjoint.

Bayes' rule, $P(C|X) \propto P(X|C)P(C)$, can be used when the posterior probability is unknown. An assumption used to estimate the probability distribution of instance given the class, $P(X|C)$, is that instances from bags of the same class are i.i.d. random samples. However, this is a poor description for MI learning.

### 3.3. Hierarchical Distributions

As an illustrative example, let the instances be the color of image patches from the class *sea* or *desert*, and let image $k$ depict a blue sea like in Figure 2a with instances $\mathbb{X}_k$, and image $\ell$ depict a turquoise sea like in Figure 2b with instances $\mathbb{X}_\ell$. The instances are realizations from $P(X|\theta_k)$ and $P(X|\theta_\ell)$, respectively, where $\theta$ is the parameter indicating the colors. If the instance distribution were dependent only on class, then $\theta_k = \theta_\ell = \theta_{sea}$, which is clearly not the case. Instance distributions are dependent not only on class, but also on bag. The random vectors in $\mathbb{X}_k$ are i.i.d., but have a different distribution than those in $\mathbb{X}_\ell$. An important distinction between uncertain objects, whose distribution depends solely on the class label [43,44], and MI learning is that the instances of two bags from the same class are not from the same distribution.

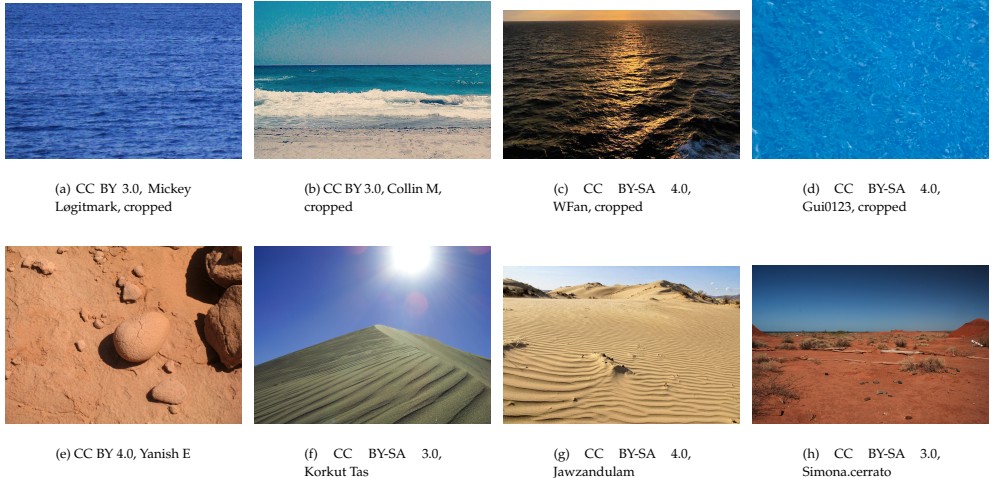

(a) CC BY 3.0, Mickey Løgitmark, cropped

(b) CC BY 3.0, Collin M, cropped

(c) CC BY-SA 4.0, WFan, cropped

(d) CC BY-SA 4.0, Gui0123, cropped

(e) CC BY 4.0, Yanish E

(f) CC BY-SA 3.0, Korkut Tas

(g) CC BY-SA 4.0, Jawzandulam

(h) CC BY-SA 3.0, Simona.cerrato

**Figure 2.** Sea and desert images from Wikimedia Commons.

The dependency nature for MI learning can be described as a hierarchical distribution (Equation (1)), where a bag, $B$, is defined as the probability distribution of its instances, $P(X|B)$, and the bag space, $\mathcal{B}$, is a set of distributions. A bag is itself a realization from the sample space of bags, whose distribution depends on the class. The generative model of instances from a positive or negative bag follows a hierarchical distribution:

$$\begin{aligned} X|B \sim P(X|B) \qquad & X|B \sim P(X|B) \\ B \sim P(B|pos) \quad \text{or} \quad & B \sim P(B|neg), \end{aligned} \tag{1}$$

respectively. From a practical viewpoint, $P(X|B)$ can be considered parametric functions, $P(X|\theta)$, where the sampling of a bag corresponds to sampling the parameter vector $\theta$ that defines its distribution:

$$X|\theta_{pos} \sim P(X|\theta_{pos}) \qquad X|\theta_{neg} \sim P(X|\theta_{neg})$$
$$\Theta_{pos} \sim P(\Theta_{pos}) \quad \text{or} \quad \Theta_{neg} \sim P(\Theta_{neg}). \tag{2}$$

The parametric generative model is shown in Figure 3.

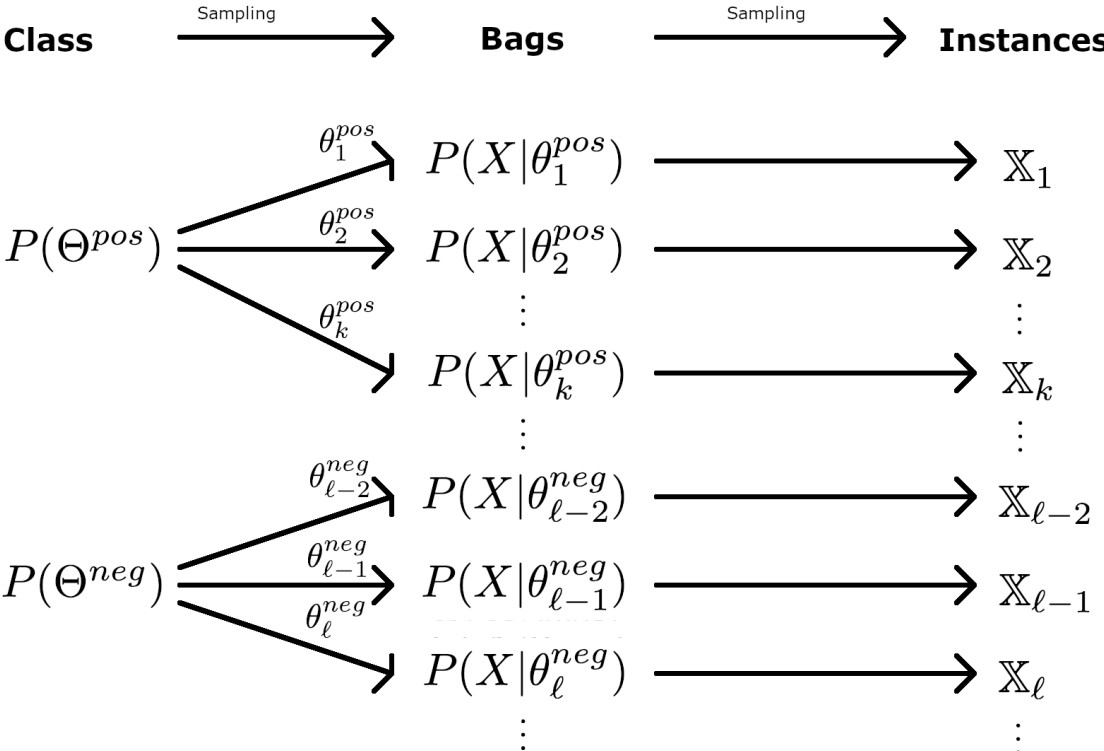

**Figure 3.** Parametric generative model. Bags are realizations of random parameter vectors, sampled according to the respective class distributions. Instances are realizations of feature vectors, sampled according the respective bag distributions. Only the instance sets are observed.

The common view in MI learning is that a bag consists of positive and negative instances, which corresponds to a bag being a mixture of a positive and a negative distribution. Consider tumor images labeled *pos* or *neg*, with instances extracted from patches. Let $P(X|\theta_k^+)$ and $P(X|\theta_k^-)$ denote the probability distributions of positive and negative segments, respectively, of image $k$. The distribution of bag $k$ is a mixture distribution:

$$P(X|\pi_k, \theta_k^+, \theta_k^-) = p_k P(X|\theta_k^+) + (1 - p_k)P(X|\theta_k^-),$$

where $p_k = \sum_{i=1}^{n_k} \tau_i / n_k$, where $\tau_i = 1$ if instance $i$ is positive. The parameter vector $(\pi_k, \theta_k^+, \theta_k^-)$ defines the bag. The probability of positive segments, $\pi_k$, depends on the image's class label, and hence $\pi_k$ is sampled from $P(\Pi_{pos})$ or $P(\Pi_{neg})$. The characteristics of positive and negative segments vary

from image to image. Hence, $\theta_k^+$ and $\theta_k^-$ are realizations of random variables, with corresponding probability distributions $P(\Theta^+)$ and $P(\Theta^-)$. The generative model of instances from a positive bag is:

$$
\begin{aligned}
X|\tau,\theta^+,\theta^- &\sim \begin{cases} P(X|\tau=1) = P(X|\theta^+) \\ P(X|\tau=0) = P(X|\theta^-) \end{cases} \\
\mathcal{T}|\pi_{pos} &\sim \begin{cases} P(\tau=1) = \pi_{pos} \\ P(\tau=0) = 1 - \pi_{pos} \end{cases} \\
\Pi_{pos} &\sim P(\Pi_{pos}), \quad \Theta^+ \sim P(\Theta^+), \quad \Theta^- \sim P(\Theta^-).
\end{aligned}
\tag{3}
$$

The corresponding sampling procedure from positive bag, $k$, is

Step 1: Draw $\pi_k$ from $P(\Pi_{pos})$, $\theta_k^+$ from $P(\Theta^+)$, and $\theta_k^-$ from $P(\Theta^-)$. These three parameters define the bag.
Step 2: For $i = 1, \ldots, n_k$, draw $\tau_i$ from $P(\mathcal{T}|\pi_k)$, draw $\mathbf{x}_i$ from $P(X|\theta_k^+)$ if $\tau_i = 1$, and from $P(X|\theta_k^-)$ otherwise.

The generative model and sampling procedure for negative bags are equivalent to that of positive bags.

By imposing restrictions, assumptions can be accurately described, e.g., the standard MI assumption: at least one positive instance in a positive bag: $P(p_k \geq 1/n_k) = 1$; no positive instances in a negative bag: $P(\Pi_{neg} = 0) = 1$; the positive and negative instance spaces are disjoint.

Equation (3) is the generative model of MI problems, assuming that the instances have unknown class labels and that the distributions are parametric. The parameters $\pi_k$, $\theta_k^+$ and $\theta_k^-$ are i.i.d. samples from their respective distributions, but are not observed and are hard to estimate due to the very nature of MI learning: the instances are not labeled. Instead, $P(X|B)$ can be estimated from the observed instances, and a divergence function can serve as classifier.

The instance i.i.d. assumption is not inherent to the probability distribution viewpoint, but the asymptotic results for the KL information discussed in Section 3.5 rely on it. In many applications, such as image analysis with sliding windows, the instances are best represented as dependent samples, but the dependencies are hard to estimate, and the independence assumption is often the best approximation. Doran and Ray [4] showed that the independence assumption is an approximation of dependent instances, but comes with the cost of slower convergence.

*3.4. Dissimilarities in MI Learning*

The information contained at bag-level is converted before it is fed into a classifier. If the bags are sets, they are commonly converted into distances. Dissimilarities in MI learning can be categorized as instance-to-instance, bag-to-bag or bag-to-class. Amores [15] implicitly assumed metricity for dissimilarity functions [27] in the BS paradigm, but there is nothing inherent to MI learning that imposes these restrictions. In the case where bags are probability distributions, distances are no longer applicable since they live in a non-vectorial space. Distances are a special case of dissimilarity functions, and the equivalent for probability distributions are referred to as divergences, $D(P_k(X), P_\ell(X))$. Although distances fulfil three properties by definition—among them symmetry and zero distance for identical sets—divergences do not have such properties, by definition.

A group of divergences named $f$-divergences has properties that are reasonable to demand for measuring the ability to distinguish probability distributions [45,46]:

*Equality and orthogonality:* An $f$-divergence takes its minimum when the two probability functions are equal and its maximum when they are orthogonal. This means that two identical bags will have minimum dissimilarity between them, and that two bags without shared sample space will have maximum dissimilarity. A definition of orthogonal distributions can be found in [47].

*Monotonicity:* The $f$-divergences possess a monotonicity property that can be thought of as an equivalent to the triangle property for distances: For a family of densities with monotone likelihood ratio, if $a < \theta_1 < \theta_2 < \theta_3 < b$, then $D(P(X|\theta_1), P(X|\theta_2)) \leq D(P(X|\theta_1), P(X|\theta_3))$. This is valid, e.g., for Gaussian densities with equal variance and mean $\theta$. This means that if the distance between $\theta_1$ and $\theta_3$ is larger than the distance between $\theta_1$ and $\theta_2$, the divergence is larger or equal. The $f$-divergences are not symmetric by definition, but some of them are.

Divergences as functions of probability distributions have not been used in MI learning, due to the lack of a probability function space defined for the BS paradigm, despite the benefit of analysis independent of specific data sets [48]. Cheplygina et al. [14] proposed using the Cauchy-Schwarz divergence with a Gaussian kernel, but as a function of the instances in the bag-to-bag setting. The KL information [29] is a non-symmetric $f$-divergence, often used in both statistics and computer science, and is defined as follows for two probability density functions (PDFs) $f_k(\mathbf{x})$ and $f_\ell(\mathbf{x})$:

$$D_{KL}(f_k, f_\ell) = \int f_k(\mathbf{x}) \log \frac{f_k(\mathbf{x})}{f_\ell(\mathbf{x})} d\mathbf{x}. \tag{4}$$

An example of a symmetric $f$-divergence is the Bhattacharyya (BH) distance, defined as

$$D_{BH}(f_k, f_\ell) = -\log \int \sqrt{f_k(\mathbf{x}) f_\ell(\mathbf{x})} d\mathbf{x}, \tag{5}$$

and can be a better choice if the absolute difference, and not the ratio, differentiates the two PDFs. The appropriate divergence for a specific task can be chosen based on identified properties, e.g., for clustering [49], or a new dissimilarity function can be proposed [50].

*3.5. Bag-to-Class Dissimilarity*

Bag-to-bag classification can be thought of as model selection: Two bags from the training set, $f_k(\mathbf{x})$ and $f_\ell(\mathbf{x})$ are the models, and unlabeled bag $f_{bag}(\mathbf{x})$ is the sample distribution, and is labeled according to which model it resembles the most. The log-ratio test is the most powerful for model selection under certain conditions (Neyman–Pearson lemma). It is possible then to perform the log-ratio test between $f_{bag}(\mathbf{x})$ and each of the bags in the training set.

The training set in MI learning is the instances, since the bag distributions are unknown. Under the assumption that the instances from each bag are i.i.d. samples, the KL information has a special role in model selection, both from the frequentist and the Bayesian perspective. Let $f_{bag}(\mathbf{x})$ be the sample distribution (unlabeled bag), and let $f_k(\mathbf{x})$ and $f_\ell(\mathbf{x})$ be two models (labeled bags). Then the expectation over $f_{bag}(\mathbf{x})$ of the log-ratio of the two models, $E\{\log(f_k(\mathbf{x})/f_\ell(\mathbf{x}))\}$, is equal to $D_{KL}(f_{bag}, f_\ell) - D_{KL}(f_{bag}, f_k)$. In other words, the log-ratio test reveals the model closest to the sampling distribution in terms of KL information [51]. From the Bayesian viewpoint, the Akaike Information Criterion (AIC) reveals the model closest to the data in terms of KL information, and is asymptotically equivalent to Bayes factor under certain assumptions [52].

An obstacles arises: The core of MI learning is that bags from the same class are not equal, e.g., two images of the sea, so that the model is most likely not in the training set. In fact, for probability distributions with continuous parameters, the probability of the new bag being in the training set is zero. For ratio-based divergences, such as the $f$-divergences, the difference between $D(f_{bag}, f_k)$ and $D(f_{bag}, f_\ell)$ becomes arbitrary. Despite their necessary properties as dissimilarity measures, and the KL information's property as most powerful model selector, we see that they can fail in practice.

If the bag sampling is sparse, the dissimilarity between $f_{bag}(\mathbf{x})$ and the labeled bags becomes somewhat arbitrary regarding the true label of $f_{bag}(\mathbf{x})$. The risk is high for ratio-based divergences such as the KL information, since $f_k(\mathbf{x})/f_\ell(\mathbf{x}) = \infty$ for $\{\mathbf{x} : f_\ell(\mathbf{x}) = 0, f_k(\mathbf{x}) > 0\}$. The bag-to-bag KL information is asymptotically the best choice of divergence function, but this is not the case for sparse training sets. Bag-to-class dissimilarity makes up for some of the sparseness by aggregation

of instances. Consider an image segment of color *deep green*, which appears in *sea* images, but not in *desert* images, and a segment of color *white*, which appears in both classes (waves and clouds). If the combination *deep green* and *white* does not appear in the training set, then a bag-to-bag KL information will result in infinite dissimilarity for all bags, regardless of class, but the bag-to-class KL information will be finite for the *sea* class.

Let $P(X|C) = \int_{\mathcal{B}} P(X|B) dP_{\mathcal{B}}(B|C)$ be the probability distribution of a random vector from the bags of class $C$. Let $D(P(X|B), P(X|pos))$ and $D(P(X|B), P(X|neg))$ be the divergences between the unlabeled bag and each of the classes. Choice of divergence is not obvious, since $P(X|B)$ is different from both $P(X|pos)$ and $P(X|neg)$, but can be done by identification of properties.

## 4. Properties for Bag-Level Classification

### 4.1. Properties for Bag-to-Class Divergences

We argue that the equality, orthogonality and monotonicity properties possessed by $f$-divergences are reasonable also for bag-to-class divergences, although less likely to occur in practice:

The equality property and the monotonicity property are valid for uncertain objects, but in practice it can occur with sparse class sampling, and we therefore argue that these properties are valid also for bag-to-class divergences. The opposite implies that a bag can be regarded more similar to one class, even though its probability distribution is identical to the other class (equality), or that, e.g., if $P_{bag}(X)$, $P_{pos}(X)$ and $P_{neg}(X)$ are Gaussian distributions with the same variance and means $\theta_{bag} < \theta_{pos} < \theta_{neg}$, we can have that $D(P(X|\theta_{bag}), P(X|\theta_{pos})) > D(P(X|\theta_{bag}), P(X|\theta_{neg}))$. In other words, we can have that the divergence between the bag and the positive class is larger than between the bag and the negative class, although the bag mean is closer to the positive class mean. This is clearly not appropriate for a dissimilarity measure.

The orthogonality property is reasonable for bag-to-class divergences: If there is no common sample space between bag and class, the divergence should take its maximum. In conclusion, $f$-divergences is the correct group for bag-to-class divergences.

There may be other desirable properties for bag-to-class divergences, where the aim is no longer to compare an i.i.d. sample to a model, but to compare an i.i.d. sample to an aggregation of models where the sample comes from one of them. We here propose two properties for bag-to-class divergences regarding infinite bag-to-class ratio and zero instance probability. Denote the divergence between an unlabeled bag and the reference distribution, $P_{ref}(X)$, by $D(P_{bag}, P_{ref})$.

In the *sea* images example, the class contains all possible colors that the sea can have, whereas a bag consists only of the colors of that particular moment in time. If the bag contains something that the class does not, e.g., brown color, this should be reflected in a larger divergence. However, the class should be allowed to contain something that the bag does not without this resulting in a similarly large divergence.

As a motivating example, consider the following: A positive bag, $P(X|a)$, is a continuous uniform distribution $\mathcal{U}(a, a + \delta)$, sampled according to $P(A) = \mathcal{U}(\eta, \zeta - \delta)$:

$$X|a \sim \mathcal{U}(a, a + \delta)$$
$$A \sim \mathcal{U}(\eta, \zeta - \delta)$$

A negative bag, $P(X|a')$, is $\mathcal{U}(a', a' + \delta')$ sampled according to $P(A') = \mathcal{U}(\eta', \zeta' - \delta')$:

$$X|a' \sim \mathcal{U}(a', a' + \delta')$$
$$A' \sim \mathcal{U}(\eta', \zeta' - \delta'),$$

and let $\eta' < \zeta$ so that there is an overlap between the two classes. For both positive and negative bags, we have that $P_{pos}(X)/P_{bag}(X) = \infty$ for a subspace of $\mathcal{X}$ and $P_{neg}(X)/P_{bag}(X) = \infty$ for a different

subspace of $\mathcal{X}$, merely reflecting that the variability in instances within a class is larger than within a bag, as illustrated in Figure 4.

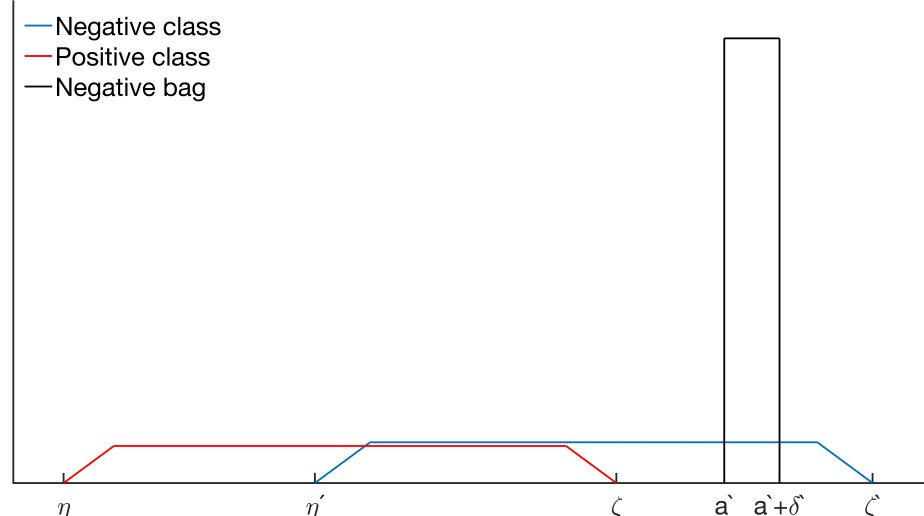

**Figure 4.** The PDF of a bag with uniform distribution and the PDFs of the two classes.

If $P_{bag}(X)$ is a sample from the negative class, and $P_{bag}(X)/P_{pos}(X) = \infty$ for some subspace of $\mathcal{X}$ it can easily be classified. From the above analysis, large bag-to-class ratio should be reflected in large divergence, whereas large class-to-bag ratio should not.

Property 1: Let $\mathcal{X}_M$ be the subspace of $\mathcal{X}$ where the bag-to-class ratio is larger than some $M$:

$$\mathcal{X}_M \subset \mathcal{X} : P_{bag}(X)/P_{ref}(X) > M,$$

and let $\mathcal{X} \setminus \mathcal{X}_M$ be its complement. Let $D^{\mathcal{X}_M}(P_{bag}, P_{ref})$ be the contribution to the total divergence for that subspace: $D(P_{bag}, P_{ref}) = D^{\mathcal{X}_M}(P_{bag}, P_{ref}) + D^{\mathcal{X} \setminus \mathcal{X}_M}(P_{bag}, P_{ref})$. Let $\mathcal{X}_M^*$ be the subspace of $\mathcal{X}$ where the class-to-bag ratio is larger than some $M$:

$$\mathcal{X}_M^* \subset \mathcal{X} : P_{ref}(X)/P_{bag}(X) > M,$$

and let $\mathcal{X} \setminus \mathcal{X}_M^*$ be its complement. Let $D^{\mathcal{X}_M^*}(P_{bag}, P_{ref})$ be the contribution to the total divergence for that subspace: $D(P_{bag}, P_{ref}) = D^{\mathcal{X}_M^*}(P_{bag}, P_{ref}) + D^{\mathcal{X} \setminus \mathcal{X}_M^*}(P_{bag}, P_{ref})$.

$D^{\mathcal{X}_M}$ approaches the maximum contribution as $M \to \infty$. $D_{\mathcal{X}_M^*}$ does not approach the maximum contribution as $M \to \infty$:

$$M \to \infty : \begin{cases} D^{\mathcal{X}_M}(P_{bag}, P_{ref}) \to \max(D^{\mathcal{X}_M}(P_{bag}, P_{ref})) \\ D^{\mathcal{X}_M^*}(P_{bag}, P_{ref}) \nrightarrow \max(D^{\mathcal{X}_M^*}(P_{bag}, P_{ref})). \end{cases}$$

Property 1 cannot be fulfilled by a symmetric divergence. This property is necessary in cases where the sample space of a bag is a subset of the sample space of the class, $\mathcal{X}_{bag} \subset \mathcal{X}_{class}$, e.g., for uniform distributions, and in cases where the variance of a bag is smaller than the variance of the class.

Consider $\mathcal{X}_M^*$. Because $P(X) < \infty$, this occurs for the subspace of $\mathcal{X}$ where $P_{bag}(X)$ is smaller than some $\epsilon$ and $P_{ref}(X)$ is not. We argue that when $P_{bag}(X) < \epsilon$, there should be no contribution to the divergence due to the very nature of MI learning: a bag is not a representation of the entire class, but only a small part of it.

Consider an unlabeled image coming from the class *sea*, and a binary classification problem with *desert* as the alternative class. If the unlabeled image contains only blue and white colors, it should not influence the divergence how the different shades of brown or green are distributed in the two classes,

as it does not influence the likelihood of this bag coming from one class or the other. This is in contrast to bag-to-bag divergences, where this indicates a bad sample-model match.

As a second motivating example, consider the same positive class as before, and the two alternative negative classes defined by:

$$A' \sim \begin{cases} P(A' = \eta') = 0.5 \\ P(A' = \eta' + 2\delta') = 0.5 \end{cases} \qquad A' \sim \begin{cases} P(A' = \eta') = 0.5 \\ P(A' = \eta' + 2\delta') = 0.25 \\ P(A' = \eta' + 3\delta') = 0.25. \end{cases}$$

For bag classification, the question becomes: from which class is a specific bag sampled? It is equally probable that a bag $P_{\eta'}(X) = P(X|A' = \eta')$ comes from each of the two negative classes, since $P_{neg}(X)$ and $P_{neg'}(X)$ only differ where $P_{\eta'}(X) = 0$, and we argue that $D(P_{\eta'}, P_{neg})$ should be equal to $D(P_{\eta'}, P_{neg'})$.

Property 2: Let $\mathcal{X}_\epsilon$ be the subspace of $\mathcal{X}$ where $P_{bag}(X)$ is larger than some $\epsilon > 0$:

$$\mathcal{X}_\epsilon \subset \mathcal{X} : P_{bag}(X) > \epsilon,$$

and let $\mathcal{X} \setminus \mathcal{X}_\epsilon$ be its complement. Let $D^{\mathcal{X}_\epsilon}(P_{bag}, P_{ref})$ be the contribution to the total divergence for that subspace: $D(P_{bag}, P_{ref}) = D^{\mathcal{X}_\epsilon}(P_{bag}, P_{ref}) + D^{\mathcal{X} \setminus \mathcal{X}_\epsilon}(P_{bag}, P_{ref})$.

The contribution to the total divergence approaches zero as $\epsilon \to 0$:

$$\epsilon \to 0 : D^{\mathcal{X}_\epsilon}(P_{bag}, P_{ref}) \to 0.$$

This property is necessary when the bag distributions are mixture distributions with possibly zero mixture proportion. It also covers the case when the bags are different distributions, not merely have different parameters, which can be modelled as a mixture of all possible distributions in the class and only one non-zero mixture proportion.

KL information is the only divergence that fulfils these two properties among the non-symmetric divergences listed in [53]. See Appendix A. As there is no complete list of divergences, it is possible that other divergences that the authors are not aware of fulfil these properties.

*4.2. A Class-Conditional Dissimilarity for MI Classification*

In the *sea* and *desert* images example, consider an unlabeled image with a *pink* segment, e.g., a boat. If *pink* is absent in the training set, then the bag-to-class KL information will be infinite for both classes. We therefore propose the following property:

Property 3: For the subspace of $\mathcal{X}$ where the alternative class probability, $P_{ref'}$, is smaller than some $\epsilon'$, the contribution to the total divergence, $D_{\mathcal{X}_{\epsilon'}}$, approaches zero as $\epsilon' \to 0$:

Let $\mathcal{X}_{\epsilon'}$ be the subspace of $\mathcal{X}$ where $P_{ref'}(X)$ is larger than some $\epsilon' > 0$:

$$\mathcal{X}_{\epsilon'} \subset \mathcal{X} : P_{ref'}(X) > \epsilon',$$

and let $\mathcal{X} \setminus \mathcal{X}_{\epsilon'}$ be its complement. Let $D^{\mathcal{X}_{\epsilon'}}(P_{bag}, P_{ref}|P_{ref'})$ be the contribution to the total divergence for that subspace: $D(P_{bag}, P_{ref}|P_{ref'}) = D^{\mathcal{X}_{\epsilon'}}(P_{bag}, P_{ref}|P_{ref'}) + D^{\mathcal{X} \setminus \mathcal{X}_{\epsilon'}}(P_{bag}, P_{ref}|P_{ref'})$.

The contribution to the total divergence approaches zero as $\epsilon' \to 0$:

$$\epsilon' \to 0 : D^{\mathcal{X}_{\epsilon'}}(P_{bag}, P_{ref}|P_{ref'}) \to 0.$$

We present a class-conditional dissimilarity that accounts for this:

$$cKL(f_{bag}, f_{pos}|f_{neg}) = \int \frac{f_{neg}(\mathbf{x})}{f_{pos}(\mathbf{x})} f_{bag}(\mathbf{x}) \log \frac{f_{bag}(\mathbf{x})}{f_{pos}(\mathbf{x})} d\mathbf{x}, \tag{6}$$

which also fulfils Properties 1 and 2, see Appendix A.

*4.3. Bag-Level Divergence Classification*

With a proper dissimilarity measure, the classification task is, in theory, straightforward: A bag is given the label of its most similar class. With dense bag and class sample, the KL bag-to-bag classifier is the most powerful. There are, however, a couple of practical obstacles: The distributions from where the instances have been drawn are not known, and must be estimated. The divergences usually do not have analytical solutions, and must therefore be approximated.

We propose two similar methods based on either the ratio of bag-to-class divergences, $rD(f_{bag}, f_{pos}, f_{neg}) = D(f_{bag}, f_{pos}))/D(f_{bag}, f_{neg})$, or the class-conditional dissimilarity in Equation (6). We propose using the KL information (Equation (4)), and report for the BH distance (Equation (5)) for comparison, but any divergence function can be applied.

Given a training set $\{(\mathbb{X}_1, y_1), \ldots, (\mathbb{X}_k, y_k), \ldots, (\mathbb{X}_K, y_K)\}$ and a set, $\mathbb{X}_{bag}$, of instances drawn from an unknown distribution, $f_{bag}(\mathbf{x})$, with unknown class label $y_{bag}$, and let $\mathbb{X}_{neg}$ denote the set of all $\mathbf{x}_{ik} \in (\mathbb{X}_k, y_k = neg)$ and $\mathbb{X}_{pos}$ denote the set of all $\mathbf{x}_{ik} \in (\mathbb{X}_k, y_k = pos)$. The bag-level divergence classification follows the steps:

1. Estimate pdfs: Fit $\hat{f}_{neg}(\mathbf{x})$ to $\mathbb{X}_{neg}$, $\hat{f}_{pos}(\mathbf{x})$ to $\mathbb{X}_{pos}$, and $\hat{f}_{bag}(\mathbf{x})$ to $\mathbb{X}_{bag}$.

2. Calculate divergences: $D(\hat{f}_{bag}, \hat{f}_{neg}))$ and $D(\hat{f}_{bag}, \hat{f}_{pos})$,

   or $cKL(\hat{f}_{bag}, \hat{f}_{pos} | \hat{f}_{neg})$ by integral approximation.

3. Classify according to: (7)

$$y_{bag} = \begin{cases} pos \text{ if } rD(\hat{f}_{bag}, \hat{f}_{pos}, \hat{f}_{neg}) < t \\ neg \text{ otherwise.} \end{cases}$$

   or

$$y_{bag} = \begin{cases} pos \text{ if } cKL(\hat{f}_{bag}, \hat{f}_{pos} | \hat{f}_{neg}) < t \\ neg \text{ otherwise.} \end{cases}$$

Common methods for PDF estimation are Gaussian mixture models (GMMs) and kernel density estimation (KDE). The integrals in step 2 are commonly approximated by importance sampling and Riemann sums. In rare cases, e.g., when the distributions are Gaussian, the divergences can be calculated directly. The threshold $t$ can be pre-defined based on, e.g., misclassification penalty and prior class probabilities, or estimated from the training set by leave-one-out cross-validation. When the feature dimension is high and the number of instances in each bag is low, PDF estimation becomes arbitrary. A solution is to estimate separate PDFs for each dimension, calculate the corresponding divergences $D_1, \ldots, D_{Dim}$, and treat them as inputs into a classifier replacing step 3.

In image analysis, it has become more and more common that MI data sets are limited by the number of (labeled) bags per class, more than the number of instances per bag. With the proposed algorithm, the PDF estimates improve with increasing number of instances, and the aggregation of class instances allows for sparser bag samples.

## 5. Experiments

*5.1. Simulated Data and Class Sparsity*

The following study exemplifies the difference between BH distance ratio, $rBH$, KL information ratio, $rKL$, and $cKL$ as classifiers for sparse training data. We investigate how the three divergences vary in accordance with the number of bags in the training set. The minimum dissimilarity bag-to-bag classifiers are also implemented, based on KL information and BH distance. The number of instances from each bag is 50, the number of bags in the training set is varied from 1 to 25 from each class,

and the number of bags in the test set is 100. Each bag and its instances are sampled as described in Equation (3), and the area under the receiver operating characteristic (ROC) curve (AUC) serves as performance measure. For simplicity, we use Gaussian distributions in one dimension for *Sim 1-Sim 4*:

$$X^- \sim \mathcal{N}(\mu^-, \sigma^{2-}) \qquad\qquad X^+ \sim \mathcal{N}(\mu^+, \sigma^{2+})$$
$$\mu^- \sim \mathcal{N}(0, 10) \qquad\qquad \mu^+ \sim \mathcal{N}(\nu^+, 10)$$
$$\sigma^{2-} = |\zeta^-|, \zeta^- \sim \mathcal{N}(1, 1) \qquad\qquad \sigma^{2+} = |\zeta^+|, \zeta^+ \sim \mathcal{N}(\eta^+, 1)$$
$$\Pi^- = \pi^- \qquad\qquad \Pi^+ = 0.10.$$

*Sim 1: $\nu^+ = 15$, $\eta^+ = 1$, $\pi^- = 0$*: No positive instances in negative bags.
*Sim 2: $\nu^+ = 15$, $\eta^+ = 1$, $\pi^- = 0.01$*: Positive instances in negative bags.
*Sim 3: $\nu^+ = 0$, $\eta^+ = 100$, $\pi^- = 0$*: Positive and negative instances have the same expectation of the mean, but unequal variance.
*Sim 4: $P(\nu^+ = 15) = P(\nu^+ = -15) = 0.5$, $\eta^+ = 1$, $\pi^- = 0.01$*: Positive instances are sampled from two distributions with unequal mean expectation.

We add *Sim 5* and *Sim 6* for the discussion on instance labels in Section 6, as follows: *Sim 5* is an uncertain object classification, where the positive bags are lognormal densities with $\mu = \log(10)$ and $\sigma^2 = 0.04$, and negative bags are Gaussian mixtures densities with $\mu_1 = 9.5$, $\mu_2 = 13.5$, $\sigma^2 = 2.5$, and $\pi_1 = 0.9$. These two densities are nearly identical, see [54], p. 15. In *Sim 6*, the parameters of *Sim 5* are i.i.d. observations from Gaussian distributions, each with $\sigma^2 = 1$ for the Gaussian mixture, and $\sigma^2 = 0.04$ for the lognormal distribution. Figure 5 shows the estimated class densities and two estimated bag densities for *Sim 2* with 10 negative bags in the training set.

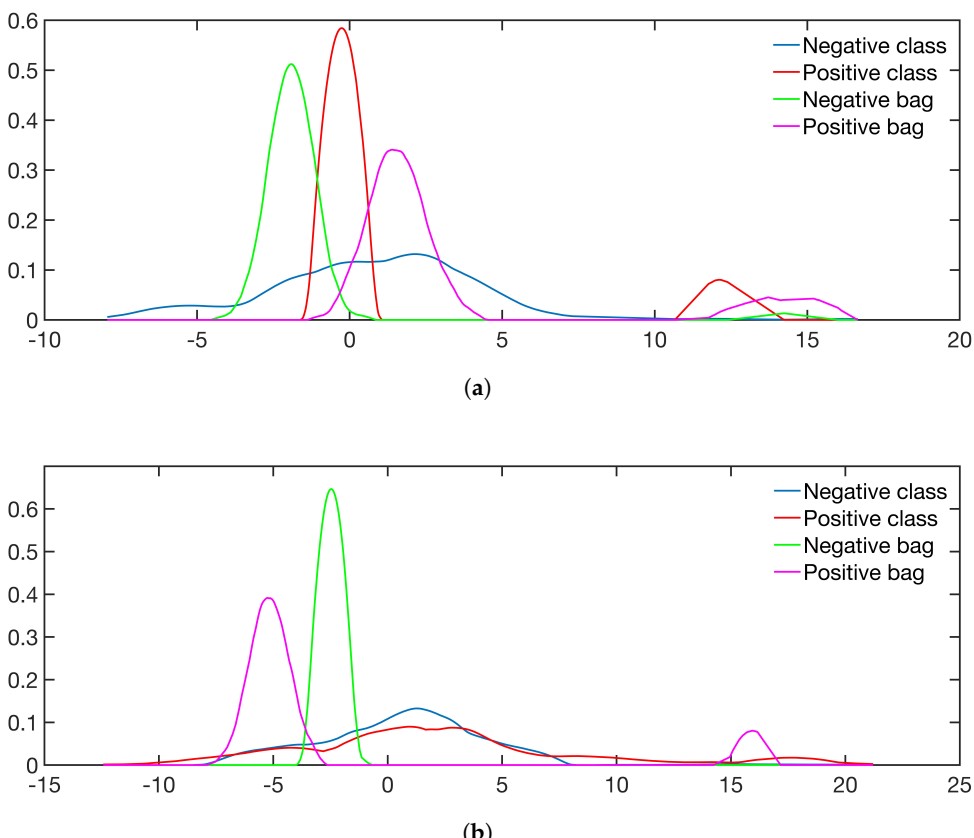

**Figure 5.** (**a**) One positive bag in the training set gives small variance for the class PDF. (**b**) Ten positive bags in the training set, and the variance has increased.

We use the following details for the algorithm in (7): 1. KDE fitting: Epanechnikov kernel with estimated bandwidth varying with the number of observations. 2. Integrals: Importance sampling. 3. Classifier: $t$ is varied to give the full range of sensitivities and specificities necessary to calculate AUC.

Table 1 shows the mean AUCs for 50 repetitions.

**Table 1.** AUC·100 for simulated data.

| | **Bags** | **neg: 5** | | | **neg: 10** | | | **neg: 25** | | |
|---|---|---|---|---|---|---|---|---|---|---|
| **Sim:** | **pos:** | **rBH** | **rKL** | **cKL** | **rBH** | **rKL** | **cKL** | **rBH** | **rKL** | **cKL** |
| | 1 | 61 | 69 | 85 | 62 | 72 | 89 | 61 | 73 | 92 |
| 1 | 5 | 63 | 75 | 86 | 64 | 82 | 94 | 68 | 84 | 97 |
| | 10 | 69 | 86 | 87 | 73 | 91 | 95 | 75 | 91 | 98 |
| | 1 | 57 | 61 | 75 | 59 | 61 | 78 | 58 | 55 | 75 |
| 2 | 5 | 59 | 67 | 79 | 60 | 68 | 84 | 62 | 63 | 85 |
| | 10 | 64 | 77 | 80 | 66 | 78 | 86 | 68 | 72 | 86 |
| | 1 | 51 | 55 | 71 | 52 | 58 | 73 | 50 | 57 | 74 |
| 3 | 5 | 53 | 61 | 76 | 53 | 66 | 81 | 52 | 65 | 83 |
| | 10 | 58 | 73 | 78 | 58 | 76 | 84 | 57 | 76 | 87 |
| | 1 | 55 | 61 | 70 | 56 | 62 | 73 | 56 | 58 | 69 |
| 4 | 5 | 56 | 63 | 75 | 57 | 64 | 81 | 59 | 59 | 80 |
| | 10 | 60 | 74 | 77 | 62 | 76 | 85 | 63 | 69 | 84 |
| | 1 | 64 | 61 | 62 | 67 | 63 | 66 | 64 | 62 | 67 |
| 5 | 5 | 73 | 69 | 63 | 74 | 70 | 67 | 75 | 71 | 72 |
| | 10 | 74 | 70 | 62 | 75 | 73 | 69 | 76 | 74 | 72 |
| | 1 | 68 | 68 | 67 | 66 | 68 | 68 | 68 | 71 | 68 |
| 6 | 5 | 65 | 64 | 67 | 68 | 68 | 69 | 70 | 71 | 74 |
| | 10 | 66 | 64 | 66 | 70 | 69 | 72 | 72 | 73 | 74 |

*5.2. The Impact of Pdf Estimation and Comparison to Other Methods*

We use a public data set from UCSB Center for Bio-Image Informatics to demonstrate the impact of PDF estimation method and for comparison with other MI classification methods. The UCSB data set consists of 58 breast tumor histology images, as seen in Figure 1). There are 32 images labeled as benign and 26 as malignant. The image patches are of size $7 \times 7$ pixels, and 708 features have been extracted from each patch. The mean number of instances per bag is 35. We have used the published instance values [14] to minimize other sources of variation than the classification algorithms. Following the procedure in [3], the principal components are used for dimension reduction, and 4-fold cross-validation is used so that $\hat{f}_{neg}(x)$ and $\hat{f}_{pos}(x)$ are fitted only to the instances in the training folds. Table 2 shows the AUC for $rKL$ and $cKL$ for three different methods for PDF estimation. GMMs are fitted to the first principal component, using an EM-algorithm, with number of components chosen by minimum AIC. In addition, KDE as in Section 5.1, and KDE with Gaussian kernel and optimal bandwidth [55] is used.

**Table 2.** AUC·100 for USCB breast tissue images.

| | **KDE (Epan.)** | **KDE (Gauss.)** | **GMMs** |
|---|---|---|---|
| *cKL* | 90 | 92 | 94 |
| *rKL* | 82 | 92 | 96 |

Table 3 shows the AUC of the GMM fitted $rKL$ and $cKL$ compared to four other MI learning methods. For articles presenting more than one method, the best-performing method is displayed in Table 3.

**Table 3.** AUC·100 for USCB breast tissue images.

| Method | AUC |
|:---:|:---:|
| *cKL* | 94 |
| *rKL* | 96 |
| DEEPISR-MIL [34] | 90 |
| Li et al. [33] | 93 |
| GPMIL [3] | 86 |
| RGPMIL [3] | 90 |

*5.3. Comparison to State-of-the-Art Methods*

The benchmark data sets that have been used for comparison of MIL methods have particularly low number of instances compared to the number of features. e.g., in *Musk*1, more than half of the bags contain less than 5 instances, and in *Musk*2, one fourth of the bags contain less than 5 instances. It is obvious that a PDF-based method will not work. The COREL data base, previously used in MIL method comparisons, is no longer available, only data sets with extracted features. Again, the number of instances is too low for density estimation. In addition, [56] showed how the feature extraction methods influence the results of MIL classifiers.

We here present the results of *cKL* and *rKL* compared to the five best-performing MIL methods using the *BreakHis* data set, as presented in [31]. This data set is suited for PDF-based methods, since the images themselves are available, and hence, the number of instances can be adjusted to assure a sufficiently dense sampling. We follow the procedure in [31], using the 162 parameter-free threshold adjacency statistics (PFTAS)

features for 1000 image patches of size $64 \times 64$. Dimension reduction is done by principal components, so that 90% of the variance is explained, and the dimension is reduced to about 25, depending on which data set, see Table 4. Each data set is split into training, validation and test sets (35%/35%/30%), where we use the exact same five test sets as [31]. There are multiple images from the same tumor, but the data set is split so that the same tumor does not appear in both training/validation and test set.

We use the following details for the algorithm in (7):

1. GMMs are fitted with $1, \ldots, 100$ components, and the number of components is chosen by minimum AIC. To save computation time, the number of components is estimated for 10 bags sampled from the training set. The median number of components is used to fit the bag PDFs in the rest of the algorithm, see Table 4. For the class PDFs, a random subsample of 10% of the instances is taken from each bag, to reduce computation time.
2. Integrals: Importance sampling.
3. Classification: To estimate the threshold, $t$, the training set is used to estimate $f_{pos}^{train}(\mathbf{x})$ and $f_{neg}^{train}(\mathbf{x})$, and the divergences between the bags in the validation set and $f_{pos}^{train}(\mathbf{x})$ and $f_{neg}^{train}(\mathbf{x})$ are calculated. The threshold, $\hat{t}$, that gives the highest accuracy will then serve as threshold for the test set.

Please note that the bags from the test set is not involved in picking the number of components or estimating $\hat{t}$.

**Table 4.** Number of components.

| Data Set | 40× | 100× | 200× | 400× |
|---|---|---|---|---|
| Dimension | 23 | 26 | 25 | 24 |
| Rep 1 | 66 | 55 | 52 | 70 |
| Rep 2 | 58 | 49 | 69 | 71 |
| Rep 3 | 59 | 50 | 50 | 70 |
| Rep 4 | 47 | 49 | 58 | 73 |
| Rep 5 | 63 | 59 | 72 | 74 |

*5.4. Results*

The general trend in Table 1 is that *cKL* gives higher AUC than *rKL*, which in turn gives higher AUC than *rBH*, in line with the divergences' properties for sparse training sets. The same trend can be seen with a Gaussian kernel and optimal bandwidth (numbers not reported). The gap between *cKL* and *rKL* narrows with larger training sets. In other words, the benefit of *cKL* increases with sparsity. This can be explained by the $\infty/\infty$ risk of *rKL*, as seen in Figure 5a. Increasing $\pi^+$ also narrows the gap between *rKL* and *cKL*, and eventually (at approximately $\pi^+ = 0.25$), *rKL* outperforms *cKL* (numbers not reported). *Sim 1* and *Sim 3* are less affected because the ratio $\pi^+/\pi^-$ is already $\infty$.

The minimum bag-to-bag classifier gives a single sensitivity-specificity outcome, and the KL information outperforms the BH distance. Compared to the ROC curve, as illustrated in Figure 6, the minimum bag-to-bag KL information classifier exceeds the bag-to-class dissimilarities only for very large training sets, typically for 500 or more, then at the expense of extensive computation time.

*Sim 5* is an example in which the absolute difference, not the ratio, differentiates the two classes, and *rBH* has the superior performance. When the extra hierarchy level is added in *Sim 6*, the performances returned to normal.

The UCSB breast tissue study shows that the simple divergence-based approach can outperform more sophisticated algorithms. *rKL* is more sensitive than *cKL* to choice of density estimation method, as shown in Table 2. *rKL* performs better than *cKL* with GMM, and both are among the best performing in Table 3. The study is too small to draw conclusions. Table 2 shows how the performance can vary between two common PDF estimation methods that do not assume a particular underlying distribution. Both KDE and GMM are sensitive to chosen parameters or parameter estimation method, bandwidth and number of components, respectively, and no method will fit all data sets. In general, KDE is faster, but more sensitive to bandwidth, whereas GMM is more stable. For bags with very few instances the benefits of GMM cannot be exploited, and KDE is preferred.

The BreakHis study shows that both *rKL* and *cKL* perform as good as or better than the other methods, the exception being *cKL* for 40×, as reported in Table 5. "As good as" refers to the mean being within one standard deviation of the highest mean. Since none of the methods have overall superior performance, we believe that the differences within one standard deviation is not enough to declare a winner. *rKL* has overall best performance in the sense that it is always within one standard deviation from the highest mean. However, *cKL*, *MI-SVM poly* and *Non-parametric* follow close behind with four out of five. Therefore, we will again avoid declaring a winner. Table 4 demonstrates that the number of components varies between repetitions, but does not influence the accuracy substantially. For reference, we have reported the AUC in Table 6, as this is a common way of reporting performance in the MIL context.

**Table 5.** Accuracy and standard deviation. Best results and those within one standard deviation in bold.

| Data Set (Magnification) | 40× | 100× | 200× | 400× |
|---|---|---|---|---|
| MI-SVM poly [57] | **86.2** (2.8) | **82.8** (4.8) | 81.7 (4.4) | **82.7** (3.8) |
| Non-parametric [58] | **87.8** (5.6) | **85.6** (4.3) | 80.8 (2.8) | 82.9 (4.1) |
| MILCNN [59] | **86.1** (4.2) | **83.8** (3.1) | 80.2 (2.6) | 80.6 (4.6) |
| CNN [31] | **85.6** (4.8) | **83.5** (3.9) | 83.1 (1.9) | 80.8 (3.0) |
| SVM [31] | 79.9 (3.7) | 77.1 (5.5) | 84.2 (1.6) | 81.2 (3.6) |
| rKL | **83.4** (4.1) | **84.9** (4.2) | **88.3** (3.6) | **84.0** (2.8) |
| cKL | 81.5 (3.2) | **85.2** (3.5) | **88.1** (3.6) | **85.0** (3.5) |

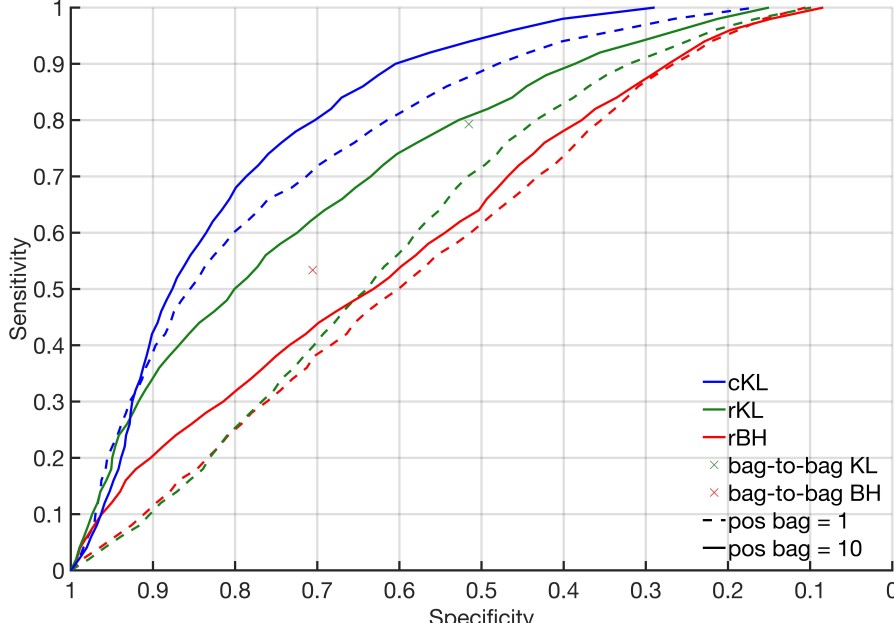

**Figure 6.** An example of ROC curves for *cKL*, *rKL* and *rBH* classifiers. The performance increases when the number of positive bags in the training set increases from 1 (dashed line) to 10 (solid line). The sensitivity-specificity pairs for the bag-to-bag KL and BH classifier is displayed for 100 positive and negative bags in the training set for comparison.

**Table 6.** AUC and standard deviation.

| Data Set (Magnification) | 40× | 100× | 200× | 400× |
|---|---|---|---|---|
| rKL | 91.4 (2.4) | 91.3 (2.2) | 94.4 (1.9) | 91.6 (1.7) |
| cKL | 88.4 (2.6) | 89.7 (1.6) | 91.9 (2.7) | 91.7 (2.4) |

The superior performance of *cKL* for the KDE (Epan.) in Table 2 can be explained by the Epanechnikov kernel's zero value, as opposed to the Gaussian kernel which is always positive. *rKL* will then suffer from its $\infty/\infty$ property given the limited training set for each class. With Gaussian kernel and GMMs, *rKL* improves its performance compared to *cKL*, as demonstrated in the simulation study. For the BreakHist data, *rKL* and *cKL* show similar performance. Although *cKL* is not within one standard deviation from the best-performing method for the 40× data set, it is within one standard deviation from *rKL*. The similar performance of *rKL* and *cKL* is in line with the simulation study where the superiority of *cKL* is demonstrated for sparse training sets, but not for all types of data.

## 6. Discussion

### 6.1. Point-of-View

The theoretical basis of the bag-to-class divergence approach relies on viewing a bag as a probability distribution, hence fitting into the branch of collective assumptions of the Foulds and Frank taxonomy [13]. The probability distribution estimation can be seen as extracting bag-level information from a set $\mathbb{X}$, and hence falls into the BS paradigm of Amores [15]. The probability distribution space is non-vectorial, different from the distance-kernel spaces in [15], and divergences are used for classification.

In practice, the evaluation points of the importance sampling gives a mapping from the set $\mathbb{X}$ to a single vector, $\hat{f}_{bag}(\mathbf{z})$. The mapping concurs with the ES paradigm, and the same applies for the graph-based methods. From that viewpoint, the bag-to-class divergence approach expands the distance branch of Foulds and Frank to include a bag-to-class category in addition to instance-level and bag-level distances. However, the importance sampling is a technicality of the algorithm. We argue that the method belongs to the BS paradigm. When the divergences are used as input to a classifier, the ES paradigm is a better description.

Carbonneau et al. [16] assume underlying instance labels. From a probability distribution viewpoint, this corresponds to posterior probabilities, which are in practice, inaccessible. In *Sim 1–Sim 4*, the instance labels are inaccessible through observations without previous knowledge about the distributions. In *Sim 6*, the instance label approach is not useful due to the similarity between the two distributions:

$$
\begin{aligned}
X|\theta^+ &\sim P(X|\theta^+) & X|\theta^- &\sim P(X|\theta^-) \\
\Theta^+ &\sim P(\Theta^+) & \Theta^- &\sim P(\Theta^-),
\end{aligned}
\tag{8}
$$

where $P(X|\Theta^+)$ and $P(X|\Theta^-)$ are the lognormal and the Gaussian mixture, respectively. Equation (3) is just a special case of Equation (8), where $\Theta^+$ is the random vector $\{\Theta, \Pi_{pos}\}$. Without knowledge about the distributions, discriminating between training sets following the generative model of Equations (3) and (8) is only possible for a limited number of problems. Even the uncertain objects of *Sim 5* are difficult to discriminate from MI objects based solely on the observations in the training set.

### 6.2. Conclusions and Future Work

Although the bag-to-bag KL information has the minimum misclassification rate, the typical bag sparseness of MI training sets is an obstacle. This is partly solved by bag-to-class dissimilarities and the proposed class-conditional KL information accounts for additional sparsity of bags.

The bag-to-class divergence approach addresses three main challenges of MI learning. (1) Aggregation of instances according to bag label and the additional class-conditioning provide a solution for the bag sparsity problem. (2) The bag-to-bag approach suffers from extensive computation time, solved by the bag-to-class approach. (3) Viewing bags as probability distributions give access to analytical tools from statistics and probability theory, and comparisons of methods can be done on a data-independent level through identification of properties. The properties presented here are not an extensive list, and any extra knowledge should be taken into account whenever available.

A more thorough analysis of the proposed function, *cKL*, will identify its weaknesses and strengths, and can lead to improved versions as well as alternative class-conditional dissimilarity measures and a more comprehensive tool.

The diversity of data types, assumptions, problem characteristics, sampling sparsity, etc. is far too large for any one approach to be sufficient. The introduction of divergences as an alternative class of dissimilarity functions, and the bag-to-class dissimilarity as an alternative to the bag-to-bag dissimilarity, has added additional tools to the MI toolbox.

**Author Contributions:** Conceptualization, K.M.; methodology, K.M.; software, K.M.; writing—original draft preparation, K.M.; writing—review and editing, J.Y.H. and F.G.; visualization, K.M.; supervision, J.Y.H. and F.G. All authors have read and agreed to the published version of the manuscript.

**Funding:** The publication charges for this article have been funded by a grant from the publication fund of UiT The Arctic University of Norway.

**Conflicts of Interest:** The authors declare no conflict of interest.

**Abbreviations**

The following abbreviations are used in this manuscript:

| | |
|---|---|
| MI | multi-instance |
| PDF | probability density function |
| IS | instance space |
| ES | embedded space |
| BS | bag space |
| KL | Kullback–Leibler |
| SVM | support vector machine |
| AIC | Akaike Information Criterion |
| GMM | Gaussian mixture models |
| KDE | kernel density estimation |
| ROC | receiver operating characteristic |
| AUC | area under the ROC curve |

**Appendix A**

For the sake of readability, we repeat summary versions of the properties here:

Property 1:

$$\mathcal{X}_M \subset \mathcal{X} : P_{bag}(X)/P_{ref}(X) > M$$

$$M \to \infty : \begin{cases} D^{\mathcal{X}_M}(P_{bag}, P_{ref}) \to \max(D^{\mathcal{X}_M}(P_{bag}, P_{ref})) \\ D^{\mathcal{X}_M^*}(P_{bag}, P_{ref}) \not\to \max(D^{\mathcal{X}_M^*}(P_{bag}, P_{ref})) \end{cases}$$

Property 2:

$$\mathcal{X}_\epsilon \subset \mathcal{X} : P_{bag}(X) > \epsilon$$

$$\epsilon \to 0 : D^{\mathcal{X}_\epsilon}(P_{bag}, P_{ref}) \to 0$$

Property 3:

$$\mathcal{X}_{\epsilon'} \subset \mathcal{X} : P_{ref'}(X) > \epsilon'$$

$$\epsilon' \to 0 : D^{\mathcal{X}_{\epsilon'}}(P_{bag}, P_{ref}|P_{ref'}) \to 0$$

*Appendix A.1. Non-Symmetric Divergences:*

We show that the only non-symmetric divergences listed in [53] that fulfil both Property 1 and Property 2 is the KL information. For all other divergences, we show one property that it does not fulfil.

The $\chi^2$-divergence, defined as:

$$\int \frac{(f_{bag}(\mathbf{x}) - f_{ref}(\mathbf{x}))^2}{f_{ref}(\mathbf{x})} d\mathbf{x},$$

does not fulfil Property 2:

$$\int_{\mathcal{X}_\epsilon} \frac{(\epsilon - f_{ref}(\mathbf{x}))^2}{f_{ref}(\mathbf{x})} d\mathbf{x} \to \int_{\mathcal{X}_\epsilon} f_{ref}(\mathbf{x}) d\mathbf{x} \not\to 0.$$

The KL information, referred to as Relative information in [53], defined as:

$$\int f_{bag}(\mathbf{x}) \log \frac{f_{bag}(\mathbf{x})}{f_{ref}(\mathbf{x})} d\mathbf{x},$$

fulfils Property 1:

$$\int_{\mathcal{X}_M} f_{bag}(\mathbf{x}) \log M \, d\mathbf{x} \to \infty = \max$$

$$\int_{\mathcal{X}_{M^*}} f_{bag}(\mathbf{x}) \log \frac{1}{M} \, d\mathbf{x} \not\to \infty,$$

since $f_{bag}(\mathbf{x}) < \infty$ and $\frac{1}{M} < \infty$, and Property 2:

$$\int_{\mathcal{X}_\epsilon} \epsilon \log \frac{\epsilon}{f_{ref}(\mathbf{x})} d\mathbf{x} \to 0 = \min$$

The Relative Jensen-Shannon divergence, defined as:

$$\int f_{bag}(\mathbf{x}) \log \frac{2 f_{bag}(\mathbf{x})}{f_{bag}(\mathbf{x}) + f_{ref}(\mathbf{x})} d\mathbf{x},$$

does not fulfil Property 1:

$$\int_{\mathcal{X}_M} f_{bag}(\mathbf{x}) \log \frac{2}{1 + \frac{f_{ref}(\mathbf{x})}{f_{bag}(\mathbf{x})}} d\mathbf{x} = \int_{\mathcal{X}_M} f_{bag}(\mathbf{x}) \log \frac{2}{1 + \frac{1}{M}} d\mathbf{x} \to \int_{\mathcal{X}_M} f_{bag}(\mathbf{x}) \log 2 \, d\mathbf{x} = \not\to \max.$$

The Relative Arithmetic-Geometric divergence, defined as:

$$\int \frac{f_{bag}(\mathbf{x}) + f_{ref}(\mathbf{x})}{2} \log \frac{f_{bag}(\mathbf{x}) + f_{ref}(\mathbf{x})}{2 f_{bag}(\mathbf{x})} d\mathbf{x},$$

does not fulfil Property 2:

$$\int_{\mathcal{X}_\epsilon} \frac{\epsilon + f_{ref}(\mathbf{x})}{2} \log \frac{\epsilon + f_{ref}(\mathbf{x})}{2\epsilon} d\mathbf{x} \to \infty \neq \min.$$

The Relative J-divergence, defined as:

$$\int (f_{bag}(\mathbf{x}) + f_{ref}(\mathbf{x})) \log \frac{f_{bag}(\mathbf{x}) + f_{ref}(\mathbf{x})}{2 f_{bag}(\mathbf{x})} d\mathbf{x},$$

does not fulfil Property 2:

$$\int_{\mathcal{X}_\epsilon} (\epsilon + f_{ref}(\mathbf{x})) \log \frac{\epsilon + f_{ref}(\mathbf{x})}{2\epsilon} d\mathbf{x} \to \infty \neq \min.$$

*Appendix A.2. Class-Conditional Bag-to-Class Divergence*

**Class-conditional KL-divergence:**

For the class-conditional divergence, there are three PDFs involved, and therefore, we have some additional restrictions. We show that the Equality and Orthogonality properties of $f$-divergences are fulfilled also by $cKL$. We were not able to conclude regarding the Monotonicity property.

$$cKL(f_{bag}, f_{pos}|f_{neg}) = \int \frac{f_{neg}(\mathbf{x})}{f_{pos}(\mathbf{x})} f_{bag}(\mathbf{x}) \log \frac{f_{bag}(\mathbf{x})}{f_{pos}(\mathbf{x})} d\mathbf{x}$$

Equality, $f_{bag}(\mathbf{x}) = f_{pos}(\mathbf{x})$, $f_{neg} \geq 0$ :

$$\int \frac{f_{neg}(\mathbf{x})}{f_{pos}(\mathbf{x})} f_{pos}(\mathbf{x}) \log \frac{f_{pos}(\mathbf{x})}{f_{pos}(\mathbf{x})} d\mathbf{x} = \int f_{neg}(\mathbf{x}) \log 1 \, d\mathbf{x} = 0 = \min.$$

Orthogonality, $f_{bag}(\mathbf{x}) / f_{pos}(\mathbf{x}) = \infty$, $f_{neg}(\mathbf{x}) > 0$ :

$$\int f_{neg}(\mathbf{x}) \frac{f_{bag}(\mathbf{x})}{f_{pos}(\mathbf{x})} \log \frac{f_{bag}(\mathbf{x})}{f_{pos}(\mathbf{x})} d\mathbf{x} = \infty = \max.$$

Property 1: $f_{neg}(\mathbf{x}) \cdot M > 0$

$$\int_{\mathcal{X}_M} f_{neg}(\mathbf{x}) M \log M \, d\mathbf{x} \to \infty = \max$$

$$\int_{\mathcal{X}_{M^*}} \frac{f_{neg}(\mathbf{x})}{M} \log \frac{1}{M} \, d\mathbf{x} \not\to \infty$$

Property 2: $\frac{f_{neg}(\mathbf{x})}{f_{pos}(\mathbf{x})} \cdot \epsilon > 0$

$$\int_{\mathcal{X}_\epsilon} \frac{f_{neg}(\mathbf{x})}{f_{pos}(\mathbf{x})} \epsilon \log \frac{\epsilon}{f_{pos}(\mathbf{x})} d\mathbf{x} \to 0$$

Property 3: $f_{bag}(\mathbf{x}) / f_{pos}(\mathbf{x}) > M$, $\epsilon' \to 0$ faster than $M \to \infty$

$$\int_{\mathcal{X}_{\epsilon'}} \frac{\epsilon'}{f_{pos}(\mathbf{x})} f_{bag}(\mathbf{x}) \log \frac{f_{bag}(\mathbf{x})}{f_{pos}(\mathbf{x})} d\mathbf{x} \to 0$$

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
