# Peer review of "A Probabilistic Bag-to-Class Approach to Multiple-Instance Learning"

_data, 2020_

Round 1

Reviewer 1 Report

In this paper, the authors proposed a multiple-instance learning for image classification. The idea is novel and good for image classification. I have the following suggestions

  1. The writing of mathematical symbols and equations must be improved. 
  2. The motivation and novelty must be emphasized in the introduction. 
  3. More details on the experiment configures should be described. 

Reviewer 2 Report

In this paper, the authors proposed bag-to-class divergence to multiple instance learning. There exist several issues in the current version.

1. The introduction section must be rewritten. It should introduce the problem, multiple instance learning, the current issue, and what the authors want to propose. The current introduction section can be merged to Section 3.

2. More visual figures should be given. For example, the authors should show a figure of the example of "sea and sky images".

3. In Table 3, the references of methods should be provided.

4. Please include the state-of-the-art methods into the comparison.

5. The performances of rKL and cKL are inconsistent. Please explain this phenomena in Table 2 and Table 3.

Reviewer 3 Report

This paper proposed a bag to class divergence based multiple instance learning algorithm which analyses the probability distribution of the instances. The classification performance of the proposed algorithm is shown on a set of synthetic and real data sets. 

This paper claims that there are four distinct contributions. But if I understand properly, the “bag-to-class dissimilarity measure” based MIL algorithm presented in the manuscript (Sec4) is the only contribution. Sec4 discusses some properties of MIL and finally build a MIL algorithm. The logical consequences of building such an algorithm are not clear from the paper.  I suggest to rewrite the paper in a way such that the reader can easily find out (i) the available theories, (ii) the intuitions developed from the available theories and (iii) the developed theory. I also suggest to provide mathematical proof if possible. 

Round 2

Reviewer 2 Report

The authors well addressed my comments.

Reviewer 3 Report

The quality of the paper is improved. It can be accepted after performing minor text editing.